# SIMPLIFYING MODELS WITH UNLABELED OUTPUT DATA

## ABSTRACT

We focus on prediction problems with high-dimensional outputs that are subject to output validity constraints, e.g. a pseudocode-to-code translation task where the code must compile. For these problems, labeled input-output pairs are expensive to obtain, but *"unlabeled"* outputs, i.e. outputs without corresponding inputs, are freely available and provide information about output validity (e.g. code on GitHub). In this paper, we present *predict-and-denoise*, a framework that can leverage unlabeled outputs. Specifically, we first train a denoiser to map possibly invalid outputs to valid outputs using synthetic perturbations of the unlabeled outputs. Second, we train a predictor composed with this fixed denoiser. We show theoretically that for a family of functions with a high-dimensional discrete valid output space, composing with a denoiser reduces the complexity of a 2-layer ReLU network needed to represent the function and that this complexity gap can be arbitrarily large. We evaluate the framework empirically on several datasets, including image generation from attributes and pseudocode-to-code translation. On the SPoC pseudocode-to-code dataset, our framework improves the proportion of code outputs that pass all test cases by 3-5% over a baseline Transformer.

## 1 INTRODUCTION

We study problems whose outputs have validity constraints. For example, in pseudocode-to-code translation, the output code must compile. Other examples include natural language translation and molecule generation, where outputs should be grammatically correct or chemically valid, respectively. State-of-the-art models typically learn the input-output mapping from expensively-obtained labeled data Kulal et al. (2019); Vaswani et al. (2017); Méndez-Lucio et al. (2020); Senior et al. (2020), which may not contain enough examples to learn a complex validity structure on high-dimensional output spaces. However, there are often lots of *"unlabeled"* outputs—outputs without a corresponding input (e.g., GitHub has over 40 million public code repositories). How do we leverage these with a much smaller amount of labeled input-output pairs to improve accuracy and validity?

In this paper, we present *predict-and-denoise*, a framework in which we compose a *base predictor*, which maps an input to a possibly invalid output, with a *denoiser*, which maps the possibly invalid output to a valid output. We first train the denoiser on synthetic perturbations of unlabeled outputs. Second, we train the base predictor composed with the fixed denoiser on the labeled data (Figure 1 left). By factorizing into two modules, base predictor and denoiser, the framework allows the base predictor to be simpler by offloading the complexity of modeling the output validity structure to the denoiser, which has the benefit of being trained on much more data.

We aim to lay down a principled framework for using unlabeled outputs with theoretical justification for improving sample efficiency by reducing the complexity of the learned base predictor. Figure 1 (middle,right) shows a pictorial example of a staircase function where valid outputs are integers and requires a complex spline to represent. When composed with a denoiser (which rounds to the nearest integer), a simple linear base predictor can represent the staircase function. We theoretically show that our framework reduces the complexity of a 2-layer ReLU network needed to represent a family of functions on a discrete valid output set in high-dimensions. This complexity gap can be arbitrarily large depending on the stability of the target function being learned. We expect such a lower complexity function to be learnable with fewer samples, improving generalization.

Empirically, we show on image generation and two pseudocode-to-code datasets (synthetic and SPoC Kulal et al. (2019)) that predict-and-denoise improves test performance across continuous and discrete output data modalities. In image generation, our framework improves the clarity and styling of font images by learning a low-complexity base predictor to generate an abstract image while the denoiser sharpens the image. For pseudocode-to-code, we consider the more difficult full-program

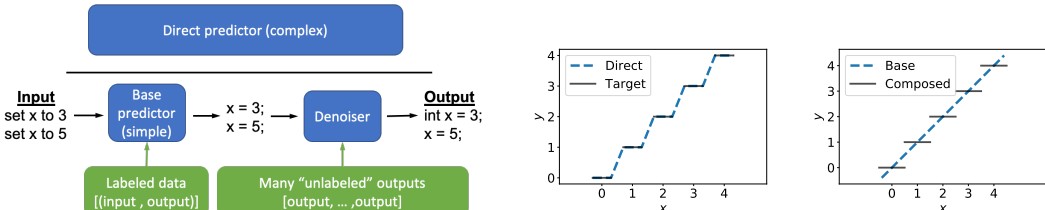

Figure 1: **(Left)** The predict-and-denoise framework: First, a denoiser is learned using synthetic perturbations of a large number of unlabeled outputs. Second, a base predictor composed with the denoiser is learned with labeled data. Composing with a denoiser allows the base predictor to be simpler, improving generalization. **(Middle)** Univariate regression example where a staircase function requires a complex linear spline fit. **(Right)** A simple linear function can fit a staircase function when composed with a denoiser which projects onto the valid outputs (the integers).

translation task rather than line-by-line translation (with compiler side information) studied by previous work Kulal et al. (2019); Yasunaga and Liang (2020). We first study a synthetic pseudocode-to-code dataset where the denoiser simplifies the base predictor by helping with global type inference. On SPOC, a recent pseudocode-to-code dataset on programming competition problems, we improve the proportion of correct programs by 3-5% points over a baseline Transformer.

## 2 SETUP

We consider prediction problems from an input space $\mathcal{X}$ (e.g., pseudocode) to an output space $\mathcal{Y}$ (e.g., code) where there is an unknown subset of *valid* outputs $\mathcal{V} \subseteq \mathcal{Y}$ (e.g., code that compiles), where the true output is always valid (in $\mathcal{V}$). We have a labeled dataset $(x_1, y_1), ..., (x_n, y_n)$ where $x_i \in \mathcal{X}$ and $y_i \in \mathcal{V}$ and access to many unlabeled outputs $(\tilde{y}_1, ..., \tilde{y}_m)$ from $\mathcal{V}$. We do not assume access to any black box function for testing validity (whether $y \in \mathcal{V}$ or not), allowing for general problems (e.g. language generation) where output validity is imprecisely characterized.

A *predictor* $f : \mathcal{X} \to \mathcal{Y}$ from a chosen hypothesis class $\mathcal{H}$ maps from inputs to the ambient output space. Our goal is to improve the predictor by leveraging information about the valid space $\mathcal{V}$ from the unlabeled examples $\{\tilde{y}_i\}_{i=1}^m$. We leverage a *denoiser* $\Pi : \mathcal{Y} \to \mathcal{V}$, which projects a possibly invalid output in $\mathcal{Y}$ and to the valid set $\mathcal{V}$. We can use unlabeled outputs to learn an approximate denoiser.

**Base, composed, and direct predictors.** Let $\|\cdot\|$ be a norm on $\mathcal{H}$. Let $\Pi \circ f_{\text{base}}$ be a *composed predictor* that is supposed to represent the target function $f^\star$ (that is, $\Pi \circ f_{\text{base}} = f^\star$ on $\mathcal{X}$). In the context of a composed predictor, we call $f_{\text{base}}$ the *base predictor*. We compare against $f_{\text{direct}} \in \operatorname{argmin}_{f \in \mathcal{H}} \{\|f\| : f(x) = f^\star(x), x \in \mathcal{X}\}$, a minimum norm *direct predictor* which represents $f^\star$.

## 3 DENOISERS CAN REDUCE MODEL COMPLEXITY

In this section, we study direct and composed predictors from an approximation standpoint and use complexity measures on predictors as surrogates for sample complexity. We aim to represent a target function $f^\star : \mathcal{X} \to \mathcal{V}$. We assume access to a denoiser $\Pi : \mathcal{Y} \to \mathcal{V}$ which projects to the nearest valid output for an appropriate metric on the output space (breaking ties arbitrarily). In Section 3.1, we give a simple example for when composing with a denoiser ($\Pi \circ f_{\text{base}}$) can drastically reduce the complexity of the learned predictor. Since $f_{\text{base}}$ becomes easier to approximate, we may expect better generalization Bartlett et al. (2017); Neyshabur et al. (2017); Wei and Ma (2020; 2019). In Section 3.2, we theoretically show for two-layer ReLU networks that the complexity required to directly represent $f^\star$ can be arbitrarily larger than representing with a composed predictor depending on the stability of $f^\star$.

### 3.1 MOTIVATING EXAMPLE

Figure 1 shows a staircase function $f^\star$ that requires a complex direct predictor $f_{\text{direct}}$ but the minimum norm base predictor $f^*_{\text{base}}$ has low complexity. For $0 < \delta < 1$, let the input space $\mathcal{X} = \biguplus_{i=1}^N [i - (1-\delta)/2, i + (1-\delta)/2]$ be a union of $N$ disjoint intervals and the valid outputs $\mathcal{V} = \mathbb{Z}$ be the integers, a subset of the ambient output space $\mathcal{Y} = \mathbb{R}$. The staircase function is $f^\star(x) = \lfloor x \rceil$ defined on $\mathcal{X}$, which rounds a linear function onto the integers. Following Savarese et al. (2019), we define the norm of a univariate function $f : \mathbb{R} \to \mathbb{R}$ as

$$\|f\| = \frac{1}{2}\max\left(\int_{-\infty}^{\infty} |f''(x)|^2 dx, |f'(-\infty) + f'(+\infty)|\right). \quad (1)$$

This norm measures the (lack of) stability of $f$. Complex functions will have a higher norm.

Consider representing $f^\star$ with linear splines, a family of piecewise linear functions. In linear splines, the norm in Equation (1) becomes roughly the sum of absolute changes in slope between piecewise segments. If we represent $f^\star$ directly with a linear spline $f_{\text{direct}}$, the norm of $f_{\text{direct}}$ has to be large due to the large number of slope changes: $\|f_{\text{direct}}\| = (N-1)/\delta$ (Figure 1 left).

Suppose we have access to a denoiser $\Pi(y) = \lfloor y \rceil$, which projects onto $\mathcal{V} = \mathbb{Z}$. Then a linear function $f^*_{\text{base}}$ composed with $\Pi$ can represent the staircase on $\mathcal{X}$, reducing the norm to 1 (Figure 1 right). By not requiring $f^*_{\text{base}}$ to represent the local complexity and discreteness in $f^\star$, the base predictor $f^*_{\text{base}}$ better captures the underlying globally linear structure of $f^\star$.

## 3.2 ANALYSIS FOR 2-LAYER RELU NETWORKS

We extend to more general hypothesis classes and high dimensional outputs. Our setting is motivated by the task of generating images of font characters from attributes, which we study empirically in Section 5.1. In font image generation, there is a discrete set of valid font images in the continuous ambient output space. Formally, we take the valid set $\mathcal{V} = \{y^*_1, ..., y^*_N\}$ to be a discrete set over $N$ output values in $\mathbb{R}^k$ and $f^\star$ is a piecewise constant function defined on $N$ disjoint intervals $\mathcal{X} = \uplus_{i=1}^N [x^l_i, x^u_i]$ (in ascending order), where there is a $\delta > 0$ gap between each interval and the next. The target function $f^\star$ is defined such that if $x \in [x^l_i, x^u_i]$, then $f^\star(x) = y^*_i$.

We study 2-layer ReLU networks, often studied as a first step towards understanding the expressivity of neural networks Neyshabur et al. (2014); Savarese et al. (2019); Eldan and Shamir (2016). Following Savarese et al. (2019), we define $f_\theta \in \mathcal{H}$ as

$$f_\theta(x) = \sum_{l=1}^h w^{(2)}_l \left[ \langle w^{(1)}_l, x \rangle + b^{(1)}_l \right]_+ + b^{(2)}_l$$

on $x \in \mathbb{R}^d$, where we will take $d = 1$ throughout. Here, $[x]_+ = \max(x, 0)$ is the element-wise ReLU nonlinearity. The parameters $\theta$ contain the hidden unit size $h \in \mathbb{N}$ and all weights and biases. We let $W^{(1)} \in \mathbb{R}^{h \times d}$ denote the matrix with $w^{(1)}_l \in \mathbb{R}^d$ as rows and let $b^{(1)}, b^{(2)}, w^{(2)} \in \mathbb{R}^h$ be vectors with $b^{(1)}_l, b^{(2)}_l, w^{(2)}_l \in \mathbb{R}$ as elements respectively. We let $\Theta$ denote this parameter space.

**Measure of complexity.** Following Savarese et al. (2019), the complexity of a network is associated with the squared Euclidean norm of the weights

$$C(\theta) = \frac{1}{2}(\|w^{(2)}\|^2_2 + \|W^{(1)}\|^2_F).$$

The norm of $f \in \mathcal{H}$ is the minimum norm required to represent $f$:

$$\|f\| = \inf_{\hat\theta \in \Theta} C(\hat\theta) \text{ s.t. } f_{\hat\theta} = f. \tag{2}$$

Savarese et al. (2019) showed that this norm is equivalent to Equation 1 for univariate networks. Since these complexity measures typically appear in generalization bounds Bartlett et al. (2017); Neyshabur et al. (2017), we expect to improve generalization error by reducing these complexity measures.

**Minimum complexity reduces with a denoiser.** Given $\Pi(y) \in \arg\min_{y^* \in \mathcal{V}} \|y^* - y\|_2$ which is projection onto $\mathcal{V}$ (breaking ties arbitrarily), we want to compare the norms of $f_{\text{direct}}$ that represents $f^\star$ directly and the minimum norm base predictor that represents $f^\star$:

$$f^*_{\text{base}} = \arg\min_{f \in \mathcal{H}} \{\|f\| : \Pi \circ f(x) = f^\star(x), x \in \mathcal{X}\}. \tag{3}$$

Note that $\|f^*_{\text{base}}\| \leq \|f_{\text{direct}}\|$ since $f_{\text{direct}}$ is a feasible solution. Thus composing cannot increase the norm.

**Adjacent intervals measure stability.** Our result depends crucially on the number of *non-adjacent* pairs of intervals in $f^\star$. Suppose the output dimension is $k = 1$. We define a pair of interval indices $(i, i+1)$ as *adjacent* if there is no valid output value $y \in \mathcal{V}$ such that either $y^*_i < y < y^*_{i+1}$ or $y^*_{i+1} < y < y^*_i$ hold. The number of non-adjacent interval pairs characterizes the instability of $f^\star$. Let $|J|$ be the number of non-adjacent pairs and $|I|$ be the number of adjacent pairs, where $|I| + |J| = N - 1$. Our bound also depends on $L = \min_i |y^*_i - y^*_{i+1}|$ and $U = \max_i |y^*_i - y^*_{i+1}|$, the min and max separation between valid points. For higher output dimensions ($k > 1$), let $y^*_{i,j}$ be the $j$-th output coordinate of the $i$-th valid point and let $|J_j|, |I_j|, L_j, U_j$ be the analogous quantities for each output coordinate $j \in [k]$.

```
1                                       1   int main() {                      1   int main() {
2   instantiate var_7;                  2     string var_7;                   2     string var_7;
3   read var_7 from stdin;              3     cin >> var_7;                   3     cin >> var_7;
4   instantiate var_5;                  4     int var_5;                      4     bool var_5;
5   read var_5 from stdin;              5     cin >> var_5;                   5     cin >> var_5;
6   set var_9 to 34;                    6     int var_9 = 34;                 6     int var_9 = 34;
7   set var_9 to 10 plus var_9;         7     var_9 += 10;                    7     var_9 += 10;
8   add "str_0" to the end of var_7;    8     var_7 += "str_0";               8     var_7 += "str_0";
9   set var_5 to max of var_5 and var_9; 9    var_5 = max(var_5, var_9);      9     var_5 = max(var_5, var_9);
10  print var_7;                        10    cout << var_7;                  10    cout var_7;
11  output var_5;                       11    cout << var_5;                  11    cout << var_5;
12  print var_9;                        12    cout << var_9;                  12    cout << var_9;
                                        13    return 0; }                     13    return 0; }
```

Figure 2: **(Left-Middle)** Example pseudocode and code from the synthetic dataset. Since the pseudocode is ambiguous, variable types and whether to instantiate a variable must be inferred. **(Right)** Random corruption used to train a denoiser from corrupted to valid code. The denoiser must infer the correct type of `var_5` from other lines.

**Theorem 1.** *Let the valid output space $\mathcal{V} \subset \mathbb{R}^k$ be a set over $N$ multivariate output values $\{y_1^*,...,y_N^*\}$ in $\mathcal{V}$. Let $f^\star : \mathbb{R} \to \mathbb{R}^k$ be a piecewise constant function defined on $\mathcal{X} = \uplus_{i=1}^N [x_i^l, x_i^u]$ where $f^\star(x) = y_i^*$ if $x \in [x_i^l, x_i^u]$. Let $\Delta_x$ be the length of the smallest interval in $\mathcal{X}$. Then*

$$\frac{\|f_{direct}\|}{\|f_{base}^*\|} = \Omega\left( \frac{N \max_j L_j}{\sum_{j=1}^k U_j\left(|J_j| + \delta \frac{|I_j|}{\Delta_x}\right)} \right) \tag{4}$$

See Appendix A for a proof. If $|J_j|$ are sublinear in $N$ and valid points are evenly spaced, then the gap is $\Omega(1/\delta)$ which can be arbitrarily large for a fixed output dimension as $\delta \to 0$ and $N \to \infty$. If any $|J_j|$ is linear in $N$ (many non-adjacent intervals), then there is only a constant factor gap in the worst case. Overall, if $f^\star$ is stable with respect to its discrete output space, we can learn a simpler base predictor that still represents $f^\star$ when composed with the denoiser. Note that in practice, we need to regularize the base predictor to find this low complexity solution.

## 4 PREDICT-AND-DENOISE FRAMEWORK

In Section 3, we assumed access to a denoiser $\Pi$ that maps output $y \in \mathcal{Y}$ to a valid output $\Pi(y) \in \mathcal{V}$, allowing the min-norm base predictor $f_{\text{base}}^*$ to have much lower complexity. In this section, we are not given a denoiser but instead have access to a large number of unlabeled outputs $\tilde{y}_1,...,\tilde{y}_m \in \mathcal{V}$. We present *predict-and-denoise*, a framework for utilizing unlabeled output examples to simplify models. In this framework, we first use self-supervised learning on the unlabeled outputs to learn an approximate denoiser $\Pi$, and then use $\Pi$ (which is now fixed) to learn a composed predictor $\Pi \circ f_\theta$. Here, $f_\theta$ is the learned base predictor with parameters $\theta$.

Figure 2 (left-middle) gives an example input-output pair in a pseudocode-to-code task. Using the predict-and-denoise framework, the model could learn to make code translations on a mostly local, line-by-line basis (a simpler solution) while relying on the denoiser to correct types globally.

**Learning the denoiser.** Assume that as domain knowledge, we have a noising distribution $q(\tilde{y}' \,|\, \tilde{y})$ over outputs given a valid output $\tilde{y}$. Figure 2 (middle) gives an example of an output program in a pseudocode-to-code translation task. Here, a noising distribution may make random semantic and syntactic corruptions such as changing types or removing semicolons and parentheses (Figure 2 right). The denoising objective here is to recover the original code from corrupted code. More generally, given the noising distribution, we train a probabilistic model $p_\beta(\tilde{y} \,|\, \tilde{y}')$ on output pairs $(\tilde{y}', \tilde{y})$ where $\tilde{y}' \sim q(\cdot \,|\, \tilde{y})$. We train the probabilistic model by maximizing the log-likelihood

$$\text{maximize}_\beta \; \mathbb{E}_{\tilde{y}}[\mathbb{E}_{\tilde{y}' \sim q}[\log p_\beta(\tilde{y} \,|\, \tilde{y}')]] \tag{5}$$

using unlabeled output samples. The denoiser $\Pi_\beta(\tilde{y}') = \text{argmax}_{\tilde{y}} \, p_\beta(\tilde{y} \,|\, \tilde{y}')$ is defined via the probabilistic model.

**Learning the composed predictor.** In this step, we fix the learned denoiser $\Pi_\beta$ and learn the composed predictor $\Pi_\beta \circ f_\theta$ on labeled examples. We train a probabilistic model $p_\theta$ for the base predictor by optimizing

$$\text{maximize}_\theta \; \mathbb{E}_{x,y}[\mathbb{E}_{y' \sim p_\theta}[\log p_\beta(y \,|\, y')]] + \lambda \mathbb{E}_{x,y}[\log p_\theta(y \,|\, x)]. \tag{6}$$

The first term maximizes a lower bound on the log-likelihood of the composed predictor via $p_\beta$ and $p_\theta$ (see Appendix D). We optimize a lower bound since optimizing the log-likelihood directly requires computing an intractable partition function over the high-dimensional output space. The second term is the log-likelihood of only $p_\theta$. We define the base predictor $f_\theta(x) = \text{argmax}_y p_\theta(y \,|\, x)$.

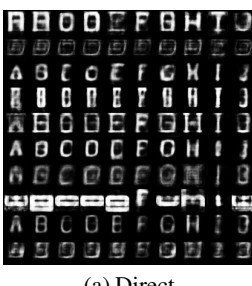 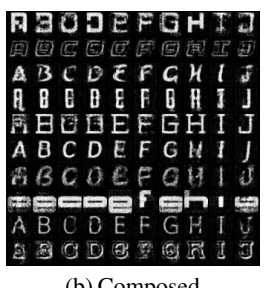 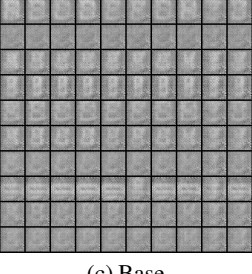

        (a) Direct               (b) Composed               (c) Base

Figure 3: Generated letters A-J for 10 randomly selected fonts. **(a)** The direct predictor makes blurry outputs with many artifacts. **(b)** The composed predictor (base + denoiser) makes clearer outputs with more distinct font patterns. **(c)** The improvement comes from leveraging output structure learned by the denoiser. This allows the base predictor to produce blurrier outputs corresponding to a lower norm model.

Since the learned $\Pi_\beta$ is imperfect, the hyperparameter $\lambda$ in the objective trades off between fitting the composition $\Pi_\beta \circ f_\theta$ and fitting $f_\theta$ directly to the data. For discrete output spaces, the first term in this objective involves an expectation over a discrete space of outputs. Depending on the model and the task, optimizing this objective may require REINFORCE Williams (1992) or a Gumbel-softmax reparameterization Jang et al. (2017); Maddison et al. (2016). The direct predictor is only trained with the second term of our objective $\mathbb{E}_{x,y}[\log p_\theta(y \,|\, x)]$.

**Choice of noising distribution.** Learning the base predictor composed with the denoiser allows for some distribution mismatch between the errors of the base predictor and the noising distribution the denoiser is trained on. By learning in a composed manner, the base predictor can adapt to the choice of noising distribution. In our experiments in Appendix B, we find that predict-and-denoise gives gains across a variety of noising distributions.

## 5 EXPERIMENTS

We evaluate predict-and-denoise on image generation from given attributes and full-program psuedocode-to-code translation, showing its benefits on both continuous and discrete output spaces. In image generation, composed models generate clearer images with less artifacts with few labeled examples. For *full-program* pseudocode-to-code translation in SPOC Kulal et al. (2019), a recent pseudocode-code dataset, our framework improves the proportion of correctly generated programs by 3-5% points over a baseline Transformer and achieves comparable or better results to a line-by-line translation model from previous work Kulal et al. (2019).

### 5.1 IMAGE GENERATION FROM ATTRIBUTES

We evaluate predict-and-denoise on font image generation, where the ambient output space is continuous. This task closely mirrors the theoretical setup, where the input is low-dimensional (index of the font and character type) to a high-dimensional output (image). We also validate the theory from Section 3, which suggested that regularization is required to realize the complexity reduction of the minimum-norm base predictor $f_{\text{base}}^*$. Qualitatively, image samples from our composed predictor are clearer and has less artifacts.

**Prediction task and denoising objective.** We mapping two one-hot vectors corresponding to the character identity (out of 62 possible) and the font of the character to generate (out of 100 fonts) $32 \times 32$ grayscale font images. Here, valid font images have cleanly defined lines and adhere to the font styling. We train using the pixel-wise squared error loss for all models and tune L2 regularization strength on a validation set. To train the composed predictor, we set $\lambda = 0$ in (6), using only the composed loss. The denoising objective is to sharpen unlabeled font images distorted by a Gaussian blur filter with randomly sampled radii in $[0,2]$. We also report gains with other noising functions (embossing, contrast perturbations) in Appendix B.

**Data.** We use a dataset of 56k fonts originally scraped from the Internet Bernhardsson (2016). Out of the 6200 labeled examples (62 characters $\times$ 100 fonts), we split randomly into 2500 training examples, 100 validation examples, and 3600 test examples. The training examples contain a random subset of the characters for each font. The models must generate the unseen characters of each font with the correct font styling at test-time. The denoiser uses additional unlabeled images for $\sim$50k other fonts.

| | Test MSE |
|---|---|
| Direct | 0.193 |
| Composed | **0.171** |

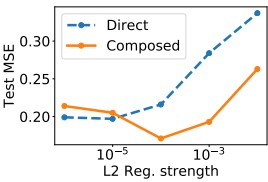

Figure 4: Test MSE on font image generation. **(Left)** Results when L2 regularization strength is tuned with the validation set. **(Right)** Varying L2 regularization strength (1e-6 to 1e-2) for direct and composed predictors. While similar at low regularization, increasing the regularization strength improves the composed predictor while hurting the direct predictor.

**Models and metrics.** The base predictor $f_\theta$ and the direct predictor $f_{\text{direct}}$ are both 7-layer fully-connected networks (see Appendix B). The denoiser $\Pi_\beta$ is a 3-layer U-Net Ronneberger et al. (2015). We test image sample quality directly by computing the pixel-wise squared error with respect to ground truth test images.

**Results.** For regularization strength tuned on the validation set (Figure 4 left), the composed predictor achieves an 11% reduction in test MSE compared to the best direct predictor test error. The direct predictor test MSE increases when its outputs are processed by the denoiser at test time. We visualize the predicted images for some randomly-selected fonts for comparison (Figure 3). The base predictor trained with L2 regularization outputs noisy gray images, suggesting that it has learned a lower complexity model. In contrast, L2 regularization does not improve the direct predictor (Figure 4 right) since directly outputting clearly defined lines and transitions between black and white pixels requires a relatively high complexity model. Note that we study L2 regularization as motivated by theory, but we expect any reasonable regularization method to help. Indeed, we find that adding dropout to the composed model improves the MSE further to 0.165. Additional results on varying labeled and unlabeled data size are in Appendix B, where the performance of the Composed model improves upon Direct on all instances.

## 5.2 PSEUDOCODE-TO-CODE

We evaluate predict-and-denoise on pseudocode-to-code translation, where the ambient output space is discrete. We evaluate on two pseudocode-to-code datasets (synthetic in Section 5.2.1 and SPOC in Section 5.2.2. On SPOC, our framework improves the proportion of programs that pass all test cases by 3-5% points over a baseline Transformer and has competitive or better results to line-by-line models Kulal et al. (2019).

**Prediction task and denoising objective.** We consider full-program pseudocode-to-code translation, where inputs $\mathcal{X}$ are human-generated pseudocode. The ambient output space $\mathcal{Y}$ is all possible strings and the set of valid outputs $\mathcal{V}$ are strings that compile with the g++ compiler. In contrast to previous works which decompose the problem into line-by-line translation and use information from the compiler Kulal et al. (2019); Yasunaga and Liang (2020), we translate the entire program at once without compiler access. Following Yasunaga and Liang (2020), the denoising objective for both pseudocode-to-code datasets consists of repairing random semantic and syntactic corruptions of unlabeled code examples (see Appendix E).

**Models and regularization.** We use a Transformer Vaswani et al. (2017) for both the base predictor and the denoiser. In all models, we use a combination of weight decay, dropout, attention dropout, and ReLU dropout as regularization. To train the composed predictor, we use $\lambda = 1$ to balance between the fitting the composed and direct objectives. During inference, we use a greedy decoding for simplicity (without beam search). Problem-specific optimizations such as beam search and querying a compiler during inference can improve the results further.

**Pretraining models.** In machine translation, a standard way to incorporate unlabeled outputs is to pretrain the encoder/decoder on monolingual data Ramachandran et al. (2018); Skorokhodov et al. (2018); Devlin et al. (2019). We consider a pretrained predictor which is pretrained with the denoising objective on unlabeled code and then trained on labeled examples, utilizing a shared encoder/decoder vocabulary. We employ predict-and-denoise on top by initializing from the pretrained model (**Pretrained+Composed**), which provides complementary benefits beyond pretraining.

**Back-translation models.** Back-translation methods use an output to input model (learned on the labeled data) applied on unlabeled outputs to generate additional synthetic inputs Sennrich et al. (2016b). We employ predict-and-denoise on top by initializing from a back-translation model (**Back-translation + Composed**), showing complementary benefits.

|  | Compile Err | Exec Err | Correct |
|---|---|---|---|
| Direct | 51.4 | 12.0 | 36.6 |
| Composed | 19.2 | 12.8 | 68.0 |
| Pretrained | 11.2 | 10.8 | 78.0 |
| Pretrained + Composed | 9.0 | 4.8 | **87.6** |
| Backtranslation | 0.8 | 10.0 | 89.2 |
| Backtranslation + Composed | 1.2 | 7.0 | **91.8** |

Table 1: Results on synthetic pseudocode-to-code task. Proportion of generated code (%) resulting in a compilation error, execution-time error, or correct code.

|  | Compile Err | Exec Err | Correct |
|---|---|---|---|
| SPOC TESTP |  |  |  |
| Direct | 75.5 | 12.3 | 12.2 |
| Direct-scaled | 75.3 | 13.4 | 11.2 |
| Composed | 74.6 | 10.2 | 15.2 |
| Pretrained | 75.3 | 10.0 | 14.8 |
| Pretrained + Composed | 74.2 | 10.4 | **15.4** |
| Backtranslation | 72.0 | 11.8 | 16.3 |
| Backtranslation + Composed | 71.2 | 11.1 | **17.7** |
| Line-by-line ref. Kulal et al. (2019) | - | - | 17.8 |
| SPOC TESTW |  |  |  |
| Direct | 48.7 | 16.6 | 34.5 |
| Direct-scaled | 48.0 | 18.5 | 33.6 |
| Composed | 47.1 | 16.2 | 36.8 |
| Pretrained | 47.5 | 15.4 | 37.0 |
| Pretrained + Composed | 46.1 | 15.8 | **38.1** |
| Backtranslation | 43.5 | 15.8 | 40.8 |
| Backtranslation + Composed | 42.4 | 15.5 | **42.1** |
| Line-by-line ref. Kulal et al. (2019) | - | - | 30.7 |

Table 2: Results on SPOC pseudocode-to-code test sets, showing proportion of outcomes of generated code (%). Direct-scaled has the same number of total layers as the composed model, but roughly double the number of trainable parameters.

```
1  set var_8 to "str_2";
2  instantiate var_2;
3  read var_2 from stdin;
4  add "str_4" to the beginning of var_8;
5  set var_2 to false;
6  set var_5 to true;
7  if var_2 is true, ←
      swap the values of var_2 and var_5;
8  if var_2 is true←
      , set var_2 to the value of var_5←
      and var_5 to the value of var_2;
9  output var_8 to stdout;
10 print var_2;
11 output var_5 to stdout;
```

```
1  int main () {
2    string var_8 = "str_2";
3    bool var_2;
4    cin >> var_2;
5    var_8 = "str_4" + var_8;
6    var_2 = false;
7    var_5 = true;
8    if ( var_2 ) {
9      string temp = var_2;
10     var_2 = var_5;
11     var_5 = temp; }
12   if ( var_2 ) {
13     bool temp = var_2;
14     var_2 = var_5;
15     var_5 = temp; }
16   cout << var_8;
17   cout << var_2;
18   cout << var_5;
19   return 0; }
```

```
1  int main () {
2    string var_8 = "str_2";
3    bool var_2;
4    cin >> var_2;
5    var_8 = "str_4" + var_8;
6    var_2 = false;
7    bool var_5 = true;
8    if ( var_2 ) {
9      bool temp = var_2;
10     var_2 = var_5;
11     var_5 = temp; }
12   if ( var_2 ) {
13     bool temp = var_2;
14     var_2 = var_5;
15     var_5 = temp; }
16   cout << var_8;
17   cout << var_2;
18   cout << var_5;
19   return 0; }
```

Figure 5: **(Left-Middle)** Example input and output of the base predictor on the synthetic dataset. **(Right)** Output of the denoiser, which instantiates `var_5` and corrects the type of `temp`.

**Metrics.** A generated program has three possible outcomes: compilation error, execution error, or correct. A program is *correct* if, executed on a set of input test cases, its outputs match the set of gold outputs. We measure the proportion of programs that fall into these outcomes.

### 5.2.1 SYNTHETIC DATASET

Pseudocode specifies local information but there are global consistency constraints to enforce (Figure 2). Modeling everything directly requires a complex model. With predict-and-denoise, the base predictor $f_\theta$ can do local translation while the denoiser $\Pi$ enforces global constraints such as type correctness. To test this intuition, we generate a synthetic pseudocode-to-code dataset where the pseudocode specifies all but the declaration types (see Figure 2).

**Dataset generation.** The synthetic programs involve 1-4 variables (bools, ints, and strings) drawn from 10 possible variable names, which are first initialized (by reading stdin) and then processed by up to 1-5 random operations, including 3 unary operations per type and 2 binary operations on ints. There are 100 possible integer values and 10 possible string values. We generate 1000 labeled examples and 20000 unlabeled code examples.

**Results.** Table 1 shows the results for all models. The Pretrained+Composed predictor improves the proportion of correct programs over direct training by 51% and over pretraining by 9.6%. We can also apply the learned denoiser to the outputs of the direct and pretrained predictors at test time, which reduces the improvement to 29.6% and 7.6% respectively. Similarly, Backtranslation+Composed improves upon direct training by 55.2% and over the strong backtranslation baseline by 2.6%. The Composed model without combining with pretraining or backtranslation still achieves a 31.4% increase over the direct model, but requires combining with pretraining or backtranslation to achieve the best performance. This suggests that predict-and-denoise offers a complementary benefit from using unlabeled output data. Results on varying unlabeled and labeled data sizes, where the composed model improves over the baselines in all instances, are in Appendix C. Figure 5 gives an example input with the output of the base and composed predictors. With the denoiser, the base predictor does not have to output all the correct variable types. Here, the denoiser correctly instantiates `var_5` and corrects the type of `temp`.

|                          | Compile Err | Exec Err | Correct |
|--------------------------|-------------|----------|---------|
| Direct-scaled            | 42.4        | 19.4     | 38.2    |
| Pretrained-scaled        | 40.2        | 19.0     | 40.8    |
| Backtranslation-scaled   | 0.2         | 17.2     | 82.6    |
| Direct + denoiser        | 23.8        | 18.2     | 58.0    |
| Pretrained + denoiser    | 9.4         | 10.6     | 80.0    |
| Backtranslation + denoiser | 1.2       | 10.2     | 88.6    |
| Pretrained + Composed    | 9.0         | 4.8      | **87.6** |
| Backtranslation + Composed | 1.2       | 7.0      | **91.8** |

Table 3: Results of scaled-up baselines and baselines with a denoiser on synthetic pseudocode-to-code task. Proportion of generated code (%) resulting in a compilation error, execution-time error, or correct code. Composed models are copied from Table 1.

### 5.2.2 SPOC

Finally, we evaluate on the challenging SPOC pseudocode-to-code dataset Kulal et al. (2019), which contains code scraped from `codeforces.com` and pseudocode written by crowdsourced workers. Since we consider the full-program translation task instead of line-by-line as in previous works Kulal et al. (2019); Yasunaga and Liang (2020), we filter out training examples where the code is longer than 1000 tokens after pre-processing, retaining over 95% (11355/11925) of the training examples. We use the two given SPOC test sets, TESTP and TESTW. TESTP tests for generalization to unseen problems, while TESTW tests for generalization to pseudocode written by different workers. We report results on the full (unfiltered) test sets.

**Denoising objective.** We use random syntactic and semantic corruptions of additional $\sim$280k unlabeled code examples from `codeforces.com` as in Yasunaga and Liang (2020). Previous program repair works Yasunaga and Liang (2020) utilize compiler error messages to guide the repair model. We only use code as input, and thus the task is relatively difficult. We define $p_\beta$ in two parts. First, we train a binary classifier $g_\gamma : \mathcal{Y} \to \{0,1\}$ which detects if a program has an error (error is label 1), trained using the denoising dataset. For an output $y'$, if $g_\gamma(y') = 0$ then we define $p_\beta(y \,|\, y') = \delta(y')$ to be a delta distribution on $y'$. Otherwise, if $g_\gamma(y') = 1$, then $p_\beta(y \,|\, y') = p_\nu(y \,|\, y')$, where $p_\nu$ is a Transformer. The Transformer $p_\nu$ is first pretrained using a linewise code repair dataset generated from unlabeled examples, then trained on full-program repair where the input program has one random corrupted line with probability 0.75. Thus, taking $\beta = (\gamma, \nu)$, we have $\Pi_\beta(y') = y'$ if $g_\gamma(y') = 0$ and $\Pi_\beta(y') = \operatorname{argmax}_y p_\nu(y \,|\, y')$ otherwise.

**Results.** On both test sets, predict-and-denoise (composed) models improve the proportion of correct code over the direct predictor by 3-5%, and applying predict-and-denoise to pretrained and backtranslation models improve them by about 1-2% (Table 2). Predict-and-denoise without pretraining or backtranslation still improved over pretraining, but improving over backtranslation requires the combining the complementary benefits of backtranslation and predict-and-denoise. Applying the denoiser to the direct and pretrained models during test time did not improve their performance. Backtranslation+Composed matches the top-1 performance of a line-by-line LSTM with attention-based copying Kulal et al. (2019) on TESTP and improve upon it by 11.4% on TESTW despite considering the more difficult full-program generation task.

### 5.3 COMPARISONS TO SCALED-UP BASELINES

Although the composed model does not optimize the denoiser during joint training with the base predictor, the final composed model consists of roughly double the number of layers as the baselines. Thus, we also present baseline results with the same number of layers as the composed model. We note that these scaled-up baselines have roughly double the amount of trainable parameters as the composed model, since the denoiser is fixed when training on labeled data.

Table 3 shows the results of scaled-up direct, pretrained, and backtranslation baselines on the synthetic code task. While the scaled direct predictor improves with respect to the unscaled direct predictor, scaling up worsens the pretrained and backtranslation baselines. Intruigingly, the pretraining becomes dramatically less effective with a very large model, possibly due to the relatively large size of the model in comparison to the unlabeled data (20k examples).

Table 2 gives results for a scaled-up direct predictor for the SPOC dataset. We find that while the compilation error rate decreases, scaling up slightly degrades the correct rate of its output programs.

### 5.4 COMPARISONS TO BASELINES WITH A DENOISER

We also give comparisons to the baselines when their outputs are post-processed by the same denoiser used by the composed model. This results in baselines with the exact same architecture and same

number of trainable parameters. The only difference is that the Composed model trains the base predictor jointly with the fixed denoiser.

Table 3 shows the results of the baselines with a denoiser. While the improvement between the composed models and the baselines decreases, predict-and-denoise gives gains of 29.6%, 7.6%, and 3.2% above direct+denoiser, pretrained+denoiser, and backtranslation+denoiser respectively. We find that using the denoiser does not improve SPoC baseline results in general. These experiments highlight the importance of training jointly with the fixed denoiser.

# 6 RELATED WORK

**Semi-supervised learning.** Like semi-supervised learning, predict-and-denoise leverages large amounts of unlabeled data. However, semi-supervised learning works typically use unlabeled *input* data Tarvainen and Valpola (2017); Miyato et al. (2018); Shu et al. (2018); Berthelot et al. (2019), whereas we have "unlabeled" outputs. In classification, unlabeled outputs can help with handling label shift Lipton et al. (2018); Azizzadenesheli et al. (2019), but otherwise there is very little output structure. If both unlabeled inputs and outputs are available, our method is complementary with semi-supervised methods.

**Denoising autoencoding.** Denoising autoencoders (DAE) are classical building blocks for unsupervised deep representation learning Vincent et al. (2008; 2010). Recently, DAEs have been considered on the input side to combat adversarial robustness by attempting to clean the adversarial example first using invariances learned from unlabeled data Gu and Rigazio (2015); Wong and Kolter (2020). We consider DAEs for learning invariances and structure in the output space instead of inputs.

**Machine translation.** Machine translation methods use monolingual data in both the source and target languages to improve their models Sennrich et al. (2016b); Cheng et al. (2016). Pretraining methods use language modeling on monolingual data to initialize the encoder and decoder Ramachandran et al. (2018); Skorokhodov et al. (2018); Devlin et al. (2019). Back-translation methods generate additional synthetic parallel examples by training on the backwards (target to source) problem Sennrich et al. (2016b). Predict-and-denoise gives complementary gains on top of pretraining and back-translation.

**Semantic parsing and structured prediction.** Some recent semantic parsing works have explicitly provided output constraints using abstract syntax trees (AST) and enforcing type constraints Yin and Neubig (2017); Krishnamurthy et al. (2017); Xiao et al. (2016); Dong and Lapata (2016). Krishnamurthy et al. (2017) note that enforcing type constraints during training not only prevents invalid outputs but also improves generalization, supporting our results. While these methods are useful when the validity structure is known and well-defined, we focus on extracting unknown structure from unlabeled outputs. Structured prediction spans applications including speech Zhang and Wu (2013), vision Mueller (2013), and medical diagnosis Jagannatha and Yu (2016). Many approaches use graphical models (on top of neural models) for enforcing validity, e.g. HMMs and CRFs in OCR and sequence tagging Kassel (1995); Huang et al. (2015). These approaches typically require carefully engineering the graphical model to integrate with a neural component and do not consider the simplicity benefits of composition.

# 7 CONCLUSION

Many tasks in machine learning are no longer classification or regression but require generating outputs with rich structure (images, text, music, proteins, etc.), for which unpaired outputs are very common. We introduce the predict-and-denoise framework, in which we compose a predictor with a denoiser trained on unpaired outputs. Open questions include whether we can train in a more differentiable way for discrete output spaces and how to choose the best denoising objective for a given prediction task.

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
