# OpenReview forum: "Simplifying Models with Unlabeled Output Data"
_ICLR.cc/2021/Conference — Reject_

### Official Review · AnonReviewer3 · 2020-10-28
**new interesting framework to leverage unlabelled output data**

**Rating:** 6
**Confidence:** 3

**Review:**

The paper introduces a “predict-and-denoise” model for structured prediction, specifically for tasks where the output has to adhere to some constraints e.g. natural language, code etc. This framework allows leveraging of unlabelled output data to train the denoiser, which consequently allows the base predictor to be of low complexity that can potentially generalize with relatively fewer labelled data. The authors theoretically back their arguments basing their theory on a 2 layer ReLU model. The paper demonstrates the performance of this model on two tasks - font image generation, and pseudocode-to-code translation and shows improvement in performance over previous works.

+ves :

- The paper is very well written and easy to follow. The motivation and the contributions are very clear, and the experimental section is also well detailed and organized.
- To the best of my knowledge the framework of predict-and-denoise learned in a composed manner and using this framework to leverage unlabelled output data is a novel contribution of the paper.
- The authors argue that this framework allows reduced complexity of the base predictor, backed theoretically for a 2 layer ReLU network. The authors have provided a detailed proof of their argument in the supplementary material, although I have not completely verified its correctness.

Concerns :

- I believe that the experimental section currently lacks fair comparisons, especially in the task of pseudocode-to-code translation. The authors compare their method with other methods for leveraging unlabelled data such as pre-training and back-translation. The authors show that predict-and-denoise framework can be applied on top of these existing approaches, and yields consistent improvement.

  However, when comparing such combinations such as “pre-training/back-translation + composed” against pre-training/back-translation, the resulting performance is not  compared with an accordingly scaled base pre-training/back-translation model to have comparable number of parameters. With a very different number of parameters in the models being compared, it's hard to say where the performance benefit is coming from.

- While the proposed method is complementary to approaches like pre-training and back translation, it will be helpful to also include comparisons such as “composed vs pre-training”, or “composed vs back-translation”. This will give an interesting comparison among these different ways of leveraging unlabelled output data. Again proper care needs to be taken about a comparable number of parameters.

While I find the framework "predict-and-denoise" very interesting, I am not entirely convinced with its empirical performance reported in the current form and I have given my score accordingly. If the authors agree with my concerns, and can try to incorporate these changes during rebuttal, I will consider updating my score.

---

> ### Author Response · Authors · 2020-11-15
> **Comparisons to scaled-up baselines and composed model without pretraining/backtranslation**
>
> We thank the reviewer for the detailed response. We address the concerns below:
>
> * **Fair comparisons between baselines and our method**
>
>     -**During the rebuttal period, we ran the baselines with the same number of total layers as the composed model (base predictor + denoiser). We note that this is roughly double the amount of trainable parameters over the composed model, since we fix the denoiser when training the base predictor. Our method still gives similar gains above these scaled-up baselines.** The large direct predictor improves to 38.2% Correct rate compared to 36.6% in the paper, while the pretrained and backtranslation baselines actually become worse with a larger number of layers. As before, we do early stopping with a validation set while checking that the models achieve perplexity 1 by the end of training. Similarly, in the SPoC task, we ran a direct baseline model with the same number of total layers as the composed model and found that the performance did not improve compared to the smaller direct baseline, even with 3x more training epochs. Due to time constraints, we did not run the backtranslation or pretraining baseline for SPoC. We will add comparisons to the scaled-up baselines in the paper.
>
>     -**We note that in addition to the table results, we compared against the baselines with a denoiser on top and show that our method gives complementary gains. We also believe this is a fair comparison because the architectures and the number of trainable parameters are exactly the same between all models and the only difference is the training method (our method trains jointly with a *fixed* denoiser), showing that the gains are directly attributable to our method.** We will make this comparison clear and update the table to also include these numbers (these were stated in the text only). Our paper states that our method gives gains (29.6%, 7.6%, and 3.2% above direct+denoiser, pretrained+denoiser, and backtranslation+denoiser respectively for synthetic code, and all SPoC results include any benefits from using the denoiser) even in the setting where the baselines process their outputs with the same denoiser our method uses. In font image generation, the direct+denoiser model has worse test MSE (0.209) than just the direct model, showing the importance of jointly training with the (fixed) denoiser. **In general, having a fixed denoiser is useful for commonly used output spaces (such as translating into English sentences), where a pretrained denoiser can be re-used to improve many models with a small amount of joint finetuning.** We’ll make this benefit clear in the paper.
>
> * **Comparisons with the composed model without pretraining or backtranslation**
>
>     -For comparison, the composed model without initializing from the pretrained or backtranslation model on SPoC achieves 15.2% and 36.8% No Err on the TestP and TestW splits, respectively. In SPoC, composed model without pretraining or backtranslation performs over 3% better than both the scaled-up and reported direct baselines. Composed performs better than pretrained+denoiser by 0.4% on TestP and is similar to pretrained+denoiser on TestW (within 0.2%). However, without combining with the strong backtranslation baseline, the composed model does not improve over backtranslation.
> On the synthetic code task, the composed model without pretraining or backtranslation achieves 68% Correct rate, which is still a 32% improvement on the direct predictor (and 10% above direct+denoiser), but gains beyond pretraining and backtranslation require a combination in this task.
>
> - We note that the baselines + denoiser results are stronger than the scaled-up versions of the baselines. In comparison to scaled-up baselines, the composed model without pretraining or backtranslation improves by 17% over the scaled-up pretraining baseline in the synthetic code task, but requires combining with backtranslation to improve beyond backtranslation.
>
>
>
> * Please see Tables 1, 2, 3 for the bolstered baseline results, as well as additional discussion in Section 5.3 and 5.4.

---

> > ### Comment · AnonReviewer3 · 2020-11-21
> > **Response to Authors**
> >
> > Thank you for responding to my concerns and questions, running relevant experiments and modifying the paper. With the new set of analyses, the comparisons look fair now. Here are my main comments :
> >
> > - Table 1 shows that the proposed "Composed" method does not seem to be an effective way of leveraging unlabelled output data in itself, and is useful only when combined with existing techniques like Pre-training and Back Translation. I think this is a potential shortcoming of the proposed approach that should be properly acknowledged by the authors.
> >
> > - The fact that composing the models with a denoiser at test time without any joint training leads to sub-optimal gains compared to joint training approach is not too surprising. However, it definitely adds value to the joint training approach proposed by the paper.
> >
> > Overall, with my concerns addressed, and empirical evidence that the proposed approach of joint training with a denoiser can provide complementary gains over existing approaches for leveraging unlabelled output data, I have increased my score. I have kept my score to weak accept, as the gains from the proposed approach are only visible when applied on top of existing techniques like Back Translation and Pre-Training.

---

> > > ### Author Response · Authors · 2020-11-24
> > > **Thanks for the followup**
> > >
> > > Thank you for the response. We acknowledge that the composed method only achieves the best performance when combining with pretraining or backtranslation in the text-based tasks (we do not combine with other methods in the image generation task), though we note that the composed method by itself is competitive or slightly better than pretrained in the more difficult real-world SPoC task. This suggests that these methods leverage different complementary benefits from unlabeled outputs and should be used together. We'll make this clearer in the paper. We also believe this is partly due to the use of the REINFORCE estimate of the gradient in the text-based tasks, and making the process fully differentiable is an interesting future direction that should improve the performance of the framework. Finally, we'd like to note that since we employ denoising autoencoding in the framework to train the decoder as a first step, initializing the base predictor with the weights of this denoiser (pretraining + composed) in the second joint training step is essentially free.

---

### Official Review · AnonReviewer1 · 2020-10-29
**Interesting idea**

**Rating:** 6
**Confidence:** 4

**Review:**

The authors propose a more data-efficient way to train generative models with constraints on the output; specifically they evaluate on image generation and pseudocode-to-code (SPoC) tasks. They train two separate models, a “predictor” and a “denoiser”, which they then compose: the output from the “predictor” is further processed by the “denoiser”. For the SPoC task they show an improvement of 3-5% over a simple transformer baseline.

The authors suggest a simple idea to make use of unlabelled data, should it be available. They use it to perturbate the unlabelled data and use the (perturbed example, example) pairs to train a denoising model. They argue that this should theoretically simplify the task of the predictor, and show improvements on several tasks. I believe that this is an interesting idea, and practically useful in the cases where data is sparse.

However, the results that they demonstrate do not seem very strong, and I would have liked to see this technique demonstrated on more competitive tasks to better gauge how well it works. The improvement of 3-5% they state seems like a low gain over a simple baseline, that may also be achievable with other techniques.

Clearly state your recommendation (accept or reject) with one or two key reasons for this choice.

I do recommend this paper to be accepted, because it clearly presents an interesting idea.

The recommendation is a “weak accept” though, because the experimental evidence for the technique is not convincing enough to me. I would have expected significant gains on a well understood task, clearly attributable to the technique.

---

> ### Author Response · Authors · 2020-11-15
> **Additional empirical evidence for comparisons**
>
> We thank the reviewer for the feedback. We address the stated concerns below:
>
> * **Regarding the strength of the baselines, the pretraining and backtranslation baselines are strong standard baselines from machine translation**, where monolingual data is used to improve models. In MT, backtranslation (Sennrich et al. 2016), and recently pretraining with denoising autoencoders (Lewis et al. 2019) have shown strong benefits with unlabeled output data. In general, good baselines that use unlabeled output data are scarce since this is a relatively underexplored field, but we compare against the strongest standard procedures.
>
> * **R1 states that it’s unclear whether the empirical gains are directly attributable to our method. During the rebuttal period, we ran all baselines with the same total number of layers as the composed model (roughly doubling the number of layers), which doubles the amount of trainable parameters compared to the composed model.** On synthetic code, the larger direct baseline improved the direct baseline to 38.2% compared to 36.6% in the paper. The pretrained and backtranslation baselines actually become worse with a larger number of layers. In SPoC, we ran a direct baseline model with the same number of total layers as the composed model and found that the performance did not improve compared to the smaller direct baseline, even with 3x more training epochs. Due to time constraints, we did not run the backtranslation or pretraining baseline for SPoC. **The gains in the paper hold also for scaled-up baselines.**
>
> * In the paper, we also compared against these strong baselines with a denoiser on top. We believe this is also a fair comparison because the architectures and the number of trainable parameters are exactly the same between all models and the only difference is the training method (our method trains jointly with a *fixed* denoiser), showing that the gains are directly attributable to our method. We will make this comparison clear and update the table to also include these numbers (these were stated in the text only). **Our paper states that our method gives gains (29.6%, 7.6%, and 3.2% above direct+denoiser, pretrained+denoiser, and backtranslation+denoiser respectively for synthetic code, and all SPoC results include any benefits from using the denoiser) even in the setting where the baselines post-process their outputs with the same denoiser our method uses.**
>
> * **R1 states that a 3-5% gain is not very large in SPoC. In terms of effect size on SPoC, we have gains of 3-5% when the overall no-error percentage is around 12%, up to a 42% relative improvement, which we believe is a significant gain given the challenging full-program translation task**, although some of the improvement is due to the backtranslation or pretraining procedure. We also do comparably (within 0.1% on TestP) or better (11% improvement on TestW) than a line-by-line baseline which was tailored specifically to the structure of the code problem.
>
>
>
> * The bolstered baseline results have been added to Table s 1,2,3, with additional discussion in Sections 5.3, 5.4.

---

### Official Review · AnonReviewer2 · 2020-10-29
**Simple and intuitive idea**

**Rating:** 6
**Confidence:** 3

**Review:**

This paper proposes a framework for problems where the output has some validity constraints, for e.g. the output must be a valid python program that must compile. These kind of problems arise naturally in settings such as pseudocode to program, and moreover there are many more unlabelled valid programs that are easily available (e.g. on Github) than there are labelled examples - i.e. paired pseudo-code, code examples. In this case, the authors propose the following framework of predict and de-noise: 1) train a de-noiser that learns to map synthetically noised versions of the un-labelled valid examples and 2) compose a predictor on the labelled examples with this de-noiser so that end predictions belong to the space of valid programs. The idea proposed in the paper is simple and intuitive, and the authors show that this approach leads to an improvement of 3-5% on the SPOC pseudo-code to code data-set. The authors also provide some theoretical justification why such a composition is the right thing to do.

Overall I like the idea in the paper, and such an approach has been used in NLP for various problems such as spelling correction etc. Besides the predict and de-noise framework another approach that is used in machine translation is the idea of using back-translation, which in the context of pseudo-code to code would look something like this: 1) train a sequence model A such as transformer on code to pseudo-code, 2) use the trained model A to generate paired training data for the unlabelled code (obtained e.g, from Github) by out-putting pseudo-code for it, and 3) train a final model on original labelled pseudo-code to code data plus the artificially generated pseudo-code to code data. I wonder how this would compare to the proposed method?

---

> ### Author Response · Authors · 2020-11-15
> **Our method gives complementary gains to backtranslation**
>
> We thank the reviewer for the comments. We address the question about comparison to a backtranslation baseline:
>
> We compared against exactly the strong backtranslation baseline that R2 described in Table 1 and Table 2 for the pseudocode-to-code tasks. We show that our method gives complementary gains on top of backtranslation. Even when the same denoiser used in our method is used on the backtranslation baseline, our composed+backtranslation model improves upon those results by 3.2% in synthetic code and 1.4% in SPoC.

---

### Author Response · Authors · 2020-11-20
**Updated draft posted, addressing reviewer concerns**

We thank all the reviewers for their time and effort. The reviewers thought that the work was **"simple and intuitive", "interesting, and practically useful in the cases where data is sparse", and "very well-written and easy to follow"**, while providing some theoretical justification for why the "framework allows reduced complexity of the base predictor.. for a 2 layer ReLU network". To their knowledge, **"this framework to leverage unlabelled output data is a novel contribution"**.

We addressed all the concerns in the individual comments and revised the draft accordingly in Tables 1,2,3 along with discussion in Sections 5.3 and 5.4. Here is a summary of the main changes:
* **R1 asked about the strength of the baselines. We clarified that the pretraining and backtranslation baselines are strong standard baselines from machine translation** , one of the few areas that have utilized unlabeled output data.
* R3 asked about **fair comparison to baselines with the same number of layers as our model**, which has a base predictor composed with an additional (fixed) denoiser. During the rebuttal period, **we ran scaled-up baselines with the same number of layers**, and found that our method still improves similarly over these baselines. We clarified that **we also compared against baselines + the denoiser our model uses, and still improve upon these, showing the efficacy of our training method.** These results have been added to Tables 1,2,3.
* R3 wanted to see the **results of our composed model without pretraining or backtranslation.** We have run these and added them to the paper.
* We clarified a misunderstanding from R2, who asked for comparisons to backtranslation, which were already in the paper.
* **R1 asked about the 3-5% effect size on SPoC. For this challenging full-program translation task, we believe this is a significant improvement (46% relative improvement** over baseline Transformer). We also compare favorably to a line-by-line baseline tailored specifically to the structure of the code problem.

---

### Decision · Program_Chairs · 2021-01-07
**Final Decision**

**Decision:**

Reject

**Comment:**

This work is well written and easy to follow and proposes a novel framework to utilize unlabeled output data. The authors have also given a detailed proof that the denoiser reduces the required complexity of the predictor. However, ultimately the experimental results are somewhat weak and leave doubts as to how effective the approach is. More convincing experimental results such as significant improvements on a well understood task and acknowledging that the approach is mostly useful when combined with pre-training and back translation would improve the work.

Pros
- Well written.
- Technically novel approach to the problem of utilizing unlabeled output data.
- Interesting proof on the reduced complexity requirement for the predictor.

Cons:
- Experimental results are not convincing. Showing significant improvements on a well understood task would be more convincing.
- The approach is only really useful when combined with pre-training or back-translation.